# Atypical Processing of Novel Distracters in a Visual Oddball Task in Autism Spectrum Disorder

**DOI:** 10.3390/bs7040079

**Published:** 2017-11-16

**Authors:** Estate M. Sokhadze, Eva V. Lamina, Emily L. Casanova, Desmond P. Kelly, Ioan Opris, Irma Khachidze, Manuel F. Casanova

**Affiliations:** 1Department of Biomedical Sciences, University of South Carolina School of Medicine-Greenville, 200 Patewood Dr., Ste A200, Greenville, SC 29615, USA; SOKHADZE@greenvillemed.sc.edu (E.M.S.); LAMINAE@mailbox.sc.edu (E.V.L.); ECasanova@ghs.org (E.L.C.); DKelly@ghs.org (D.P.K.); 2Developmental Behavioral Unit, Department of Pediatrics, Children’s Hospital, Greenville Health System, Greenville, SC 29615, USA; 3School of Medicine, University of Miami, Miami, FL 33136, USA; ioanopris.phd@gmail.com; 4Centre of Experimental Biomedicine, 14 Gotya str., Tbilisi 0160, Georgia; irmakha@yahoo.com

**Keywords:** event-related potential, autism spectrum disorder, attention, cognitive processes, reaction time

## Abstract

Several studies have shown that children with autism spectrum disorder (ASD) show abnormalities in P3b to targets in standard oddball tasks. The present study employed a three-stimulus visual oddball task with novel distracters that analyzed event-related potentials (ERP) to both target and non-target items at frontal and parietal sites. The task tested the hypothesis that children with autism are abnormally orienting attention to distracters probably due to impaired habituation to novelty. We predicted a lower selectivity in early ERPs to target, frequent non-target, and rare distracters. We also expected delayed late ERPs in autism. The study enrolled 32 ASD and 24 typically developing (TD) children. Reaction time (RT) and accuracy were analyzed as behavioral measures, while ERPs were recorded with a dense-array EEG system. Children with ASD showed higher error rate without normative post-error RT slowing and had lower error-related negativity. Parietal P1, frontal N1, as well as P3a and P3b components were higher to novels in ASD. Augmented exogenous ERPs suggest low selectivity in pre-processing of stimuli resulting in their excessive processing at later stages. The results suggest an impaired habituation to unattended stimuli that incurs a high load at the later stages of perceptual and cognitive processing and response selection when novel distracter stimuli are differentiated from targets.

## 1. Introduction

Autism Spectrum Disorder (ASD) is characterized by severe disturbances in reciprocal social relations, varying degrees of language and communication difficulties, and behavioral patterns which are restricted, repetitive, and stereotyped [1]. Additionally, individuals with autism usually present excessive reactions to change in their environment such as aversive reactions to visual, auditory, and tactile stimuli. These perception and sensory reactivity abnormalities, found in a majority of subjects with ASD, affect their ability to effectively process information [2]. In a series of electrophysiological studies conducted by our group we explored specifics of event-related potential (i.e., ERP) reflecting information processing during performance on reaction time (RT) tasks in children with ASD [3,4,5,6,7,8,9]. Our prior studies explored the manifestations of excessive local connectivity and impaired distal functional connectivity, excessive cortical excitation/inhibition ratio, and deficient executive functioning in ASD by analyzing behavioral performance on attention tasks with concurrent dense-array ERP recording. More detailed theoretical considerations related to the results of these studies were discussed in our reviews on this topic [9,10,11].

The current study explored atypicality of reactivity to novel stimuli in autism during performance on a three-stimulus visual oddball task as reflected in RT and ERP. Elicitation of ERP waves related to novelty processing can be readily achieved through this type of oddball paradigm, in which subjects are exposed to continuous succession of three types of stimuli, one presented frequently (standard), while the two other types of stimuli are rare. One of the rare stimuli is designated as a target, whereas the second type rare stimuli has some distinction from target and is usually referred to as a novel, or novel distracter. Analysis of ERP components using this test paradigm is a useful way of investigating task-relevant and -irrelevant information processing stages as well as selective attention. Amplitude and latency characteristics of negative and positive ERP components at selected scalp topographic regions-of-interest (ROI) can provide valuable information about the early sensory perception processes and the higher-level processes including attention, cortical inhibition, response selection, error monitoring, memory update, and other cognitive activity related to working memory [11,12,13,14,15]. 

Event-related potentials reflect the activation of neural structures in primary sensory cortex, and in associative cortical areas related to higher order cognitive processes. ERP components can be categorized as short-latency (exogenous, e.g., N1) or long-latency (endogenous, e.g., P3) ERPs, which reflect early-stage, modality-specific and late-stage polymodal associative processing respectively. The early ERP components (e.g., P1, N1) reflect exogenous processes modulated by the physical attributes of the stimulus (i.e., brightness for visual stimuli; loudness of auditory stimuli, etc.), rather than by endogenous cognitive processes [16]. However, it has been noted that attention processes may operate even at the early stages of information intake and influence stimulus processing at the later stage [17]. Two endogenous components of the ERP, namely the N2 and P3, are thought to be directly associated with the cognitive processes of perception and selective attention [18]. The posterior visual N2 (labeled as N2b) is enhanced if the presented stimulus contains a perceptual feature or attributes defining the target in the task. In majority of oddball tasks, the N2b is related to the cognitive processes of stimulus identification and distinction [18,19]. In a modification of the oddball task, when rare distracters are presented along with standard and rare target stimuli, these distracters elicit a fronto-central P3a, whereas the targets elicit a parietally distributed P3b [20,21]. In a three stimulus oddball task the P3a is interpreted as “orienting” of attention to novelty, and the P3b as an index of ability to sustain attention to target. Error sensitivity can be examined by measuring response-locked ERP components associated with cortical responses to committed errors. Two ERP components relevant in this context are the error-related negativity (ERN) and the error-related positivity (Pe).

This study employed a visual novelty oddball task with simultaneous recording of motor responses and brain potentials in children with ASD and in typically developing (TD) children. Such an approach allowed us to analyze attentional and cognitive processing mechanisms recruited in typically developing subjects relative to children with autism. We proposed that early stages of the sensory visual signal processing in autism might be characterized by low selectivity and by reduced adaptation to task-irrelevant items. We predicted that in autism, as compared to TD controls, there would be a lower selectivity in early ERPs components in response to infrequent target, frequent non-target and rare novel distracters as well as delayed endogenous ERP components. In particular, we anticipated higher amplitudes of P1 and N1 not only to target but also to standard and novel stimuli reflective of deficits in stimulus category selectivity. Amplitude and latency of N2 and P3 components in ASD and TD children were expected to differ during later stages of information processing suggesting that a more effortful allocation was needed to sustain attention on target differentiation from the novel distracters. Differentiation of motor responses in terms of their correctness reflected in error-related potentials (ERN, Pe) was also expected to be deficient in children with ASD, and was predicted to negatively affect accuracy of motor responses. 

## 2. Methods and Materials

### 2.1. Participants

Participants with autism spectrum disorder (ASD) (age range 9 to 18 years) were recruited through the Weisskopf Child Evaluation Center (WCEC). Diagnosis was made according to the Diagnostic and Statistical Manual of Mental Disorders (DSM-IV-TR) [22] or DSM-5 [1] and further ascertained with the Autism Diagnostic Interview—Revised (ADI-R) [23]. They also had a medical evaluation by a developmental pediatrician. All subjects had normal hearing based on past hearing screens. Participants either had normal vision or wore corrective lenses. Participants with a history of seizure disorder, significant hearing or visual impairment, a brain abnormality conclusive from imaging studies or an identified genetic disorder were excluded. All participants were high-functioning children with ASD with full scale IQ > 80 assessed using the Wechsler Intelligence Scale for Children, Fourth Edition (WISC-IV; [24]), the Stanford-Binet Intelligence Test [25], or the Wechsler Abbreviated Scale of Intelligence (WASI, [26]). 

Typically developing (TD) children were recruited through advertisements in the local media. All TD participants were free of neurological or significant medical disorders, had normal hearing and vision, and were free of psychiatric, learning, or developmental disorders based on self- and parent reports. Subjects were screened for history of psychiatric or neurological diagnosis using the Structured Clinical Interview for DSM-IV Non-Patient Edition (SCID-NP, [27]). Participants within the control and autism groups were attempted to be matched for age, gender, and the socioeconomic status of their family. 

The mean age of 32 participants enrolled in the ASD group was 13.09 ± 2.41 years (range 9–18 years, 29 males, 3 females), while the mean age of the TD group (N = 24) was 13.91 ± 2.91 years (9–20 years, 20 males, 4 females). The age difference between groups was not significant (*p* = 0.38, n.s.). All children with autism were high functioning individuals. Mean Full Scale IQ score for children with autism was 90.1 ± 14.9 and were collected from their most recent IQ evaluation records. Most of children in TD group did not have their IQ records available and their mean IQ data was not possible to report. Socioeconomic status of ASD and control groups was compared based on parent education and annual household income. The approximate household incomes did not reveal any statistically significant group differences. Participants in both groups had similar parent education levels.

The study complied with all relevant national regulations and institutional policies and was approved by the local Institutional Review Board (IRB). Participating subjects and their parents (or legal guardians) were provided with full information about the study including the purpose, requirements, responsibilities, reimbursement, risks, benefits, alternatives, and role of the local IRB. The consent and assent forms approved by the IRB were reviewed and explained to all subjects who expressed interest to participate. All questions were answered before consent signature was requested. If the individual agreed to participate, both she/he and parent/guardian signed and dated the consent form and/or assent form and received a copy countersigned by the investigator who obtained consent. 

### 2.2. ERP Data Acquisition, and Signal Processing 

Electroencephalographic (EEG) data was acquired with a 128 channel Electrical Geodesics Inc. (EGI) system (v. 200) consisting of Geodesic Sensor Net electrodes, Net Amps and Net Station software (Electrical Geodesics Inc., Eugene, OR, USA) running on a Macintosh G4 computer. EEG data were sampled at 500 Hz and 0.1–200 Hz analog filtered. Impedances were kept under 40 KΩ. According to the Technical Manual of EGI [28] this Net Sensor electrode impedance level is sufficient for quality recording of EEG with this system. The Geodesic Sensor Net is a lightweight elastic thread structure containing Ag/AgCl electrodes housed in a synthetic sponge on a pedestal. The sponges were soaked in a KCl solution to render them conductive. EEG data was recorded continuously. EEG channel with high impedance or with visually detectable artifacts (e.g., channel drift, gross movement, etc.) were marked as bad using the Net Station event marker tools in “on-line” mode for further removal in the “off-line” mode using the Net Station Waveform Tools (NSWT). 

Stimulus-locked EEG recordings were segmented off-line into 1000 ms epochs spanning 200 ms pre-stimulus to 800 ms post-stimulus around the critical stimulus events: e.g., in an oddball task: (1) rare target, (2) rare non-target distracter (novel), (3) frequent non-target (standard). Data sets were digitally screened for artifacts (eye blinks, movements), and contaminated trials were removed using artifact rejection tools. The Net Station Waveform Tools’ Artifact Detection module in “off-line” mode marks EEG channel bad if fast average amplitude exceeds 200 μV, differential average amplitude exceeds 100 μV or if channel has zero variance. Segments were marked bad if they contained more than 10 bad channels, or if eye blink or eye movement were detected (>70 μV). After detection of bad channels, the NSWT’s “bad channel replacement” function was used for the replacement of data in bad channels with data interpolated from the remaining good channels (or segments) using spherical splines [29,30,31,32]. Response-locked ERPs were segmented off-line into 1000 ms epochs spanning 500 ms pre-error motor response to 500 ms post around the critical response event—commission error. Remaining data were digitally filtered using 60 Hz Notch and 0.3–20 Hz bandpass filters and then segmented by condition and averaged to create ERPs. Averaged ERP data were baseline corrected and re-referenced into an average reference frame. All stimuli and behavioral response collection was controlled by a PC computer running E-prime software (Psychology Software Tools Inc., Sharpsburg, PA, USA). Visual stimuli were presented on a 15 inch display. Manual responses were collected with a 5-button keypad (Serial Box, Psychology Software Tools Inc., Sharpsburg, PA, USA). 

### 2.3. Three Stimuli Visual Oddball Test with Novel Distracters

This test represented a modification of traditional visual three-stimulus oddball task. Stimuli letters “X”, “O”, and novel distracters (“v”, “^”, “>” and “<” signs) were presented on the screen after fixation mark “+”. One of the stimuli (“O”) was presented on 50% of the trials (frequent standard); the novel stimuli stimulus (e.g., “>”) was presented on 25% of the trials (rare distracter), whereas the third (“X”) was presented on the remaining 25% of the trials and represented the target. Subjects were instructed to press a key when they see the target letter on the screen. Each stimulus was presented for 250 ms, with a variable (1000–1100 ms) inter-trial interval. There were 480 trials in total, with a break every 240 trials. The complete sequence took around 20 min. 

### 2.4. Behavioral Measures 

Behavioral response measures were mean reaction time (RT in ms) and response accuracy (percent of correct hits). Number and percent of commission and omission errors along with total number of errors was calculated for each participant. Post-error RT was calculated as RT to the first correct response after committed error (either omission of commission error). Difference between post-error RT and preceding correct RT to target was used to calculate normative post-error RT slowing values [5,8,33]. Analysis of distribution of RT in correct and error trials was conducted using sigmoid curve methodology from Opris et al. [34].

### 2.5. Event-Related Potentials (ERP)

#### 2.5.1. Stimulus-Locked ERPs

ERP dependent measures were: adaptive mean amplitude and latency of ERP peak (e.g., P3a, P3b) within a temporal window across a region-of-interest (ROI) channel group. A list of ERP dependent variables included stimulus-averaged amplitude and latency of the frontal ERP components: N1 (80–180 ms post-stimulus), N2 (200–320 ms), and P3a (300–520 ms); and the posterior (parietal ROIs) ERP components P1 (80–180 ms), N2 (N2b, 180–320 ms) and P3b (320–560 ms). The frontal ROIs for N1, N2 and P3a components included following EGI channels: left ROI—EGI channel 12 (between FC1 and FCz), F1, F3 and FC1; midline ROI—FCz, Fz; right ROI—EGI channel 5 (between F2 and FCz), F2, F4 and FC2. The parietal ROI for P1, N2 and P3b components included following EGI channels: left ROI—P1, P3, P7, PO3; midline—Pz; right ROI—EGI channel P2, P4, P8, PO8.

#### 2.5.2. Response-Locked Dependent Variables (ERN/Pe)

Response locked dependent variables in this study were adaptive mean amplitude and latency of the Error-related Negativity (ERN peaking within 40–150 ms post-error) and Error-related Positivity (Pe, peaking within 100–300 ms post-error). The ROI for both ERN and Pe components included FCz, sites between FCz and FC3-C1, and between FCz and FC2-C2).

### 2.6. Statistical Data Analysis 

Statistical analyses were performed on the subject-averaged behavioral (RT, error rate) and ERP data with the subject averages being the observations. The primary analysis model was the repeated measures ANOVA, with dependent variables being reaction time (RT), accuracy, commission and omission error rate, post-error RT slowing index, RT in correct and commission error trials, response locked ERN and Pe, and all the specific stimulus-locked ERP components’ (N1, P1, N2, P3) amplitudes and latencies at the selected frontal and parietal ROIs. The data of each ERP dependent variable for each relevant ROI was analyzed using ANOVA with the following factors (all within-participants): *Stimulus* (target, novel, standard) and *Hemisphere* (left, right). The between subject factor was *Group* (ASD, CNT). Histograms with normal distribution curves along with skewness and kurtosis data were obtained for each dependent variable to determine normality of distribution and appropriateness of data for ANOVA test. In all ANOVAs, Greenhouse-Geisser corrected *p*-values were employed where appropriate. Since ASD group differences from TD group were the most important aspects of the study, the focus of analysis was on main effects of *Group* (ASD vs. TD) and their interactions with other variables. For the estimation of the effect size [35] we used Partial Eta Squared (η_p_^2^) measure. Statistical analysis was performed using SPSS v.22 and Sigma Stat 3.1 packages.

## 3. Results

### 3.1. Behavioral Responses

Reaction time to targets in autism group was not different from TD group (473.2 ± 93.5 vs. 465.6 ± 98.4, F_1,54_ = 0.08, *p* = 0.78, n.s.), but the difference in commission error rate was significant (5.57 ± 9.32 percent in ASD vs. 1.12 ± 1.20 percent in TD, F_1,54_ = 4.48, *p* = 0.040), being higher in the ASD group. The difference between mean RT in correct trials and post-error trials (i.e., post-error RT -minus- correct trial RT) was negative in autism but positive in controls, and this between-group difference was significant (−15.7 ± 50.1 ms in ASD vs. 19.4 ± 37.9 ms in TD, F_1,54_ = 6.00, *p* = 0.019, see Figure 1). Analysis of distribution of RT in correct and error trials using sigmoid curve method [34] did not show group differences, however in the ASD group distribution of RT in the error trials tended to be in the faster bin range (moda ~200 ms in error vs. ~400 ms in correct trials, see Figure 2.) 

### 3.2. Motor Response-Locked Fronto-Central ERPs

**ERN and Pe.** One subject from the ASD group and 4 subjects from the TD group had no commission errors, therefore ERN/Pe datasets were comprised of 31 autistic children and 20 TD controls. Amplitude of the midline fronto-central ERN was significantly less negative in the ASD group as compared to the TD group (−2.69 ± 6.17 μV vs. −6.36 ± 4.43 μV, F_1,49_ = 4.94, *p* = 0.031). Latency of the ERN in the ASD group was delayed (107.5 ± 36.1 ms vs. 80.1 ± 20.7 ms, F_1,49_ = 9.46, *p* = 0.003). The only group difference in Pe measure was detected for the latency, though this difference barely reached minimal significance level and was featured by the tendency for prolonged latency in the ASD group (206.4 ± 44.9 ms vs. 183.3 ± 33.3 ms, F_1,49_ = 4.05, *p* = 0.05). 

### 3.3. Stimulus-Locked Event-Related Potentials

#### 3.3.1. Frontal ERPs

**N1**. *Stimulus* (target, standard, novel) factor had large main effect on N1 amplitude in both hemispheres (F_2,53_ = 7.26, *p* = 0.002, η_p_^2^ = 0.22). Amplitude of the frontal N1 to targets was bilaterally more negative in ASD group as compared to TD group (−3.68 ± 2.07 μV vs. −1.79 ± 2.29 μV, F_1,54_ = 10.01, *p* = 0.003). The ASD group showed as well more negative N100 amplitude to standards (−4.22 ± 1.71 μV vs. −2.39 ± 2.15 μV, F_1,54_ = 12.05, *p* = 0.001) and novels (−4.74 ± 2.02 μV vs. −2.61 ± 2.09 μV, F_1,54_ = 13.87, *p* < 0.001). Figure 3 illustrates group differences in response to target and novel stimuli. There were no interactions to report. Latency of the frontal N1 did not show any statistical group differences. 

**N2a.** Most pronounced group differences resulted when target and novel stimuli effects were compared. Stimulus type had medium main effect on amplitude of the bifrontal N2 (F_2,53_ = 4.84, *p* = 0.032, η_p_^2^ = 0.09). The ASD group in response to targets showed more negative amplitude bilaterally (−3.44 ± 3.45 μV vs. −1.46 ± 2.47 μV, F = 5.22, *p* = 0.027). There was significant *Stimulus* (target, novel) X *Group* (ASD, TD) interaction (F_1,54_ = 4.82, *p* = 0.033, η_p_^2^ = 0.08), that can be described as comparable N200 amplitude to targets and novels in the ASD group, while in the TD group amplitude was larger for novel stimuli. Latency of N200 component was significantly longer in the ASD group across all 3 categories of stimulation without any hemispheric differences (targets, F_1,54_ = 9.33, *p* = 0.004; standards, F_1,54_ = 6.84, *p* = 0.012; novels, F_1,54_ = 6.96, *p* = 0.011).

**P3a (Novelty P3)**. Rare *Stimulus* (target, novel) type had moderate main effect on P3a amplitude across both hemispheres (F_2,53_ = 4.22, *p* = 0.041, η_p_^2^ = 0.08). Group differences were found only for novel distracters, in particular, the ASD group had bilaterally higher amplitude as compared to the TD group (5.03 ± 3.33 μV vs. 2.88 ± 3.96 μV, F_1,54_ = 4.44, *p* = 0.04). Children with ASD as compared to the TD children had as well higher P3a amplitude to standards (F_1,54_ = 7.56, *p* = 0.008), but amplitude of P3a did not show any group differences in response to targets. Group differences for P3a latency were found only in the left hemisphere for all three categories of stimuli, and all of them featured a more prolonged latency in the ASD group (left frontal ROI- targets, F_1,54_ = 4.88, *p* = 0.031; standards, F_1,54_ = 5.07, *p* = 0.028; novels, F_1,54_ = 4.73, *p* = 0.034, see Figure 4 and Figure 5). *Hemisphere* (left, right) X *Group* (ASD, TD) interaction was significant, and can be described as more prolonged latency of the P3a component at the left hemisphere in the ASD group (F_1,54_ = 6.73, *p* = 0.012, η_p_^2^ = 0.11). 

#### 3.3.2. Parietal and Parieto-Occipital ERPs

**P1***. Stimulus* type (target, standard, novel) had marginal bilateral main effect on amplitude of the posterior P100 component (F_2,53_ = 3.20, *p* = 0.049, η_p_^2^ = 0.05). Group differences were expressed in a higher amplitude in the ASD group to standards (3.57 ± 2.08 μV vs. 2.33 ± 2.41 μV, F_1,54_ = 4.15, *p* = 0.046) and to novels (4.24 ± 2.36 vs. 2.55 ± 2.26, F_1,54_ = 7.34, *p* = 0.009, see Figure 6). Comparison of P1 amplitude to target and novel stimuli showed moderate *Stimulus* X *Group* interaction (F_1,54_ = 4.36, *p* = 0.042, η_p_^2^ = 0.08) where the ASD group showed higher response to novels as compared to the TD group. There were found no latency group differences for this component.

**N2b.** Amplitude of the parietal N2b component did not show any group differences. On the other hand, latency of N2b was globally delayed in the ASD group to all stimuli, and this effect was more pronounced on the left ROI (targets, 257.1 ± 34.1 ms in ASD vs. 222.5 ± 41.0 ms in TD, F_1,54_ = 10.78, *p* = 0.002; standards, 253.7 ± 40.2 ms vs. 211.1 ± 59.9 ms, F_1,54_ = 9.38, *p* = 0.004; novels, 248.1 ± 35.6 ms vs. 216.5 ± 53.5 ms, F_1,54_ = 6.45, *p* = 0.014). *Stimulus* (target, novel) X *Group* interaction was significant (F_1,54_ = 6.08, *p* = 0.017, η_p_^2^ = 0.11) and this effect was featured by a prolonged latency to targets in the ASD group. 

**P3b.**
*Stimulus* type had large main effect on P3b amplitude in both hemispheres (F_2,53_ = 43.75, *p* < 0.001, η_p_^2^ = 0.62). Analysis of the parietal P3b amplitude yielded group differences only in response to novel stimulus (7.24 ± 3.02 μV in ASD vs. 4.93 ± 4.32 μV in TD, F_1,54_ = 5.63, *p* = 0.021). Stimulus factor also had strong main effect on the latency of P3b peak (F_2,53_ = 8.76, *p* < 0.001; η_p_^2^ = 0.24). However, we didn’t find any P3b latency group differences or any significant interactions. 

## 4. Discussion 

The reaction time findings in this study indicate that children with ASD had a less accurate behavioral performance. Committed motor response errors were not followed by a normative post-error slowing of reaction time. Error-related negativity was less pronounced and prolonged in the ASD group, though error-relative positivity only tended to be delayed. Both early (P1, N1) and late (P3a, P3b) ERP components in response to novel distracters were enhanced and showed larger amplitude in the ASD group when compared to the TD group. The N2 component showed delayed latency to all categories of stimuli at the frontal and parietal regions of interest. Results partially replicate our earlier ERP findings in autism. Our prior studies showed similar group differences in reaction time tasks using two different types of visual three-stimulus oddball tests, including one with illusory Kanizsa figures as the stimuli [3,4,5,6,8,9]. 

In numerous research studies and reviews autistic children have been found to differ from typical children mainly with respect to the P3 in regular oddball tasks (reviewed in Cui et al. [36]). Kemner et al. [37,38,39] reported abnormally small occipital and reduced central P3 in response to visual stimuli. At the same time, these authors also reported that the parietal N2 and P3b was larger in autistic children and interpreted their results in terms of differences in attentional resource allocation, as the parietal P3b is more sensitive to such task manipulations as stimulus relevance and probability [40] and less dependent on modality as compared to the occipital P3 [39]. In general, studies using a simple visual target detection, as compared to cross-modal (e.g., audio-visual integration) tasks have found no significant differences in the P3b to targets in children with autism compared to typical controls, while abnormalities were present in dissociations of frontal (delayed) and posterior (relatively intact) P3 in visual attention tasks [41]. It should be noted, that in most ERP studies using oddball tasks, the focus was on ERP responses to target stimuli, and less attention was paid to atypically large ERP magnitude to non-target stimuli (i.e., standards and/or novel distracters). Our results in this study, as well as in our prior research studies [3,4,5,6,7,8,9] emphasize the need to perform more detailed analysis of the specifics of excessive reactivity to task-irrelevant items in oddball tests in children with ASD. 

The finding of increased amplitude of the frontal P3a to novels in children with autism is of special interest as this ERP measure is directly related to orienting to novelty. The frontal novelty P3a is less explored in autism and results are not consistent [41,42,43]. It was reported that the frontal P300 which reflects attention orienting was delayed or missing in subjects with autism and this finding was interpreted by the authors as a disruption of both parieto-frontal and cerebello-frontal networks critical for efficient cross-modal integration. Kemner et al. [37] reported that the visual N200 to novel distracters is larger when a person with autism is performing a task even when these novel stimuli are not relevant to the task in question. Abnormalities in central sensory processing both in auditory and visual modalities have been described by different authors in autism [37,39,43,44,45]. However, most of these studies analyzed and reported findings relating P3b to targets [42,46,47] and only a few P3a to standards or novels [41,43]. Despite the large number of studies published on ERPs in autism, there are not many reports about ERP components similar to those analyzed in this study. Most of the studies outline hyperactivation as well as an abnormal pattern of primary perceptual processes (e.g., low selectivity), abnormal top-down attentional control (e.g., orienting to novelty) and irregular information integration processes [48,49]. In control subjects the frontal P3a occurs earlier and commonly precedes parietal P3b, but in autistic subjects the P3a and P3b components were found to peak almost at the same time over the frontal and parietal sites in a spatial visual attention task [41]. In our study, novel stimuli elicited a delayed P3a component in the autism group. The latency of this component usually is associated with speed of attentional orienting to stimulus and reflects prefrontal working memory processes. Even less explored are early ERPs potentials in oddball tasks in ASD. The role of exogenous components and contribution to abnormalities of behavior in autism deserves further investigation.

In our prior ERP studies [3,5] on novelty processing in ASD, we reported that children with ASD showed significantly higher amplitudes and longer latencies of early frontal ERPs and delayed latency of P3a to novel distractor stimuli. The current study replicates those results mostly in terms of larger amplitudes of early potentials (P1, N1) and augmented magnitude of late potentials (P3a, P3b) to novel distracters. Our results suggest low selectivity in modality specific pre-processing of stimulus. Higher amplitude of the early frontal negativity and parietal positivity in the autism group with minimal differentiation of response magnitude to either target or non-target stimuli is an interesting finding that replicates our previous reports [6,7,8,9] where different types of visual oddball tasks were used. The visual N1 is considered to be an index of stimulus discrimination [50]. The visual N1 generally is augmented during pre-attentional stimulus processing [51], and is larger towards task-relevant target stimuli rather than unattended stimuli [52]. The ASD group shows clearly augmented and delayed frontal P3a that might result in an impaired early differentiation of target and non-target items (e.g., on N1 stage) and more effortful compensatory strategies involved for successful target identification, and following correct motor response selection. In general, the autistic group showed prolonged latencies to standard and rare non-target cues in visual oddball task. These results suggest that individuals with autism probably over-process information needed for the successful differentiation of target and distractor stimuli. 

Cortical activity during different stages of visual information processing can be detected with ERPs, as they represent stimulus-driven corticoelectric field potentials. The early ERP components are a series of potentials that are recorded at the scalp following sensory stimulation that usually occur between 40 and 200 ms post-stimulation. These components are also characterized by being exogenous in nature (i.e., they are predictably generated by delivering sensory stimulation without a need for the subject to perform any mental operations). This characteristic differentiates the early latency ERPs from endogenous ERPs (e.g., P3b), which require the performance of a cognitive task such as a novelty task used in our study. Dysfunctional selective filtering of the stimuli may occur at the levels of P1 and N1 response related to the sensory gating stage. Sensory gating is based on selective attention concepts as attention towards one stimulus requires automatic concurrent inhibition of attention towards another one. Habituation to irrelevant sensory input is an important function for the information processing by the brain, because the failure might be associated with mental disturbances. The relationship between sensory gating and oversensitivity of individuals with autism to sensory stimulation other than auditory stimulation remains understudied and the clinical correlates of visual P100 and N100 ERP components in autism are yet to be examined in-depth. 

A number of potentially interesting correlations were found between P1 and N1 sensory gating measures and P3 variables in schizophrenia [53]. Among these correlations, the positive interdependence between the N1 and P3 (P3b) latency is of interest as it is relevant to the findings of our previous [3,5,8] and current study. The study of Boutros et al. [53] suggests that decreased gating at the N1 phase of information processing negatively impacts the speed of information processing as measured by P3b latency. The further observation that this correlation was stronger in schizophrenia patients [53] emphasizes the possible deleterious effects of abnormal P1 and N1 level gating on information processing. This finding raises the possibility that the early sensory gating deficit may also impact the resource allocation capacity of the cortex as measured by P3b amplitude. 

Belmonte and Yurgelin-Todd [48,49] suggested that perceptual filtering of incoming stimulation in autism can be considered as occurring in an all-or-none manner with low specificity for the task relevance of the stimulus. This notion assumes that filtering of perceptual items may primarily be driven by the control of general arousal level rather than the activation of only modality-specific cortical areas. According to the latter authors the attention of individuals with autism seems to be founded more on the coarse control of general arousal than on selective activation of specific perceptual systems. It is reasonable to suggest that an active inhibition of irrelevant distracters is not properly functioning and allows both task-relevant and task-irrelevant stimuli to pass through earlier filtering processes creating the overload on the later stages of stimuli processing in the context of task. It is unsurprising in this context that an increased ratio of excitation/inhibition in key neural systems and high “cortical noise” has been considered as a core abnormality of autism [54,55]. 

In this study, and similar to our studies on error monitoring in autism [7,8], we found that the ERN component of the response-locked ERP was substantially decreased in children with autism. In particular, the amplitude of ERN was less negative and the latency of ERN was prolonged in the ASD group as compared to the TD children. The ERN is an electroencephalographic measure associated with the commission of errors, thought to be independent of conscious perception [33], while the Pe is thought to reflect the motivational or emotional significance of the error or, in other words, the conscious evaluation of the error [56]. It cannot be ruled out that ERN impairments are influenced by deficits in earlier perceptual processes, or attentional and working memory processes in children with autism, that might be reflected in altered stimulus-locked early and late ERPs. It has been suggested [57,58] that both the response-locked ERN and the stimulus-locked frontal N2 might reflect similar processes (i.e., response conflict detection and monitoring) and have similar neural correlates (i.e., dipole in the ACC, see also van Veen and Carter [59]). On the behavioral level, we found no group differences in RT, and only group differences between the percentages of commission (and not omission) error in the visual novelty oddball. After an error, ASD patients did not show accuracy improvement through post-error RT slowing as typical controls did. This finding replicates our previous reports [5,6,7,8]. Normally, performance on these trials is improved as a result of a change in speed–accuracy strategy which reflects executive control functioning [60]. The atypical post-error performance of ASD children (i.e., speeding instead of normative slowing) suggests the presence of an executive control deficiency. The impairment of adaptive error-correction behavior may have important consequences in daily life as optimal error-correction is necessary for adequate behavioral responses.

As demonstrated in previous studies [61], the posterior medial frontal cortex, and more specifically the rostral ACC division, is the main brain area responsible for error processing, suggesting that ASD patients have reduced posterior medial frontal cortex functioning. This area is involved when there is a need for adjustments to achieve goals [61]. The findings pointing that children with ASD have an impaired ability to improve their response accuracy by slowing down the response speed on post-error trials agrees with this notion. However, it is necessary to take into account that observed significant group differences between ASD and typical controls are manifested not only in the behavioral performance measures on reaction time tasks and associated response monitoring indices such as ERN and Pe. Group differences were also noted in terms of amplitude and latency characteristics of early ERP components preceding motor response selection (frontal and parietal P1, N1, N2) and those reflecting context update and closure (e.g., P3b) in visual oddball task [5] and various auditory tasks [44]. The sum of the group differences across these behavioral and stimulus- and response-averaged ERP indices of the ASD patients’ performance is that it reflects global deficits in attentional processes, more specifically deficits in effective differentiation of target and novel distracter stimuli. This latter interpretation is supported by the significant differences between the ASD patients and typically developing controls in terms of both the stimulus-locked and response-locked ERP amplitudes and latencies.

Post-error adaptive correction of responses might be explained by some recent neurobiological findings. There are reports about an excessive preservation of short-distance connections (i.e., local over-connectivity) and relatively poor long-distance connections (i.e., distant under-connectivity) in the neocortex of individuals with autism [62,63,64,65]. These cortical connectivity abnormalities may explain why persons with autism tend to focus on details rather than perceiving the whole Gestalt. This over-focusing on details may imply an excessively laborious and ineffective way of handling each trial in the cognitive test, and lower availability of resources after an error when effort is needed to react appropriately. This may result in insufficient activation of the ACC [66], and thus error detection and post-error reaction may be hampered [67,68]. Structural and functional deficiencies of the ACC may contribute to the atypical development of joint attention and social cognition in autism [69]. Such interpretation of the results of the ERN/Pe deficits found in several studies [6,7,66,70] is consistent with many aspects of theory and research suggesting that ACC-mediated response monitoring may contribute to social-emotional and social-cognitive development in autism [69]. However, while emphasizing the possible role of ACC-related self-monitoring deficits in autism, Mundy [69] also noted that according to Devinsky and Luciano [71] this ACC impairment related behavioral deficits emerge only when they are combined with disturbances in other related functional neural networks, e.g., dorsolateral prefrontal cortex (DLPFC). Another factor that might affect variability of ERN is related to developmental changes, as performance monitoring in children and adolescents endures changes due to maturation processes [72], while our sample included both young children (9–12 years old range) and adolescents (i.e., teenagers in 13–18 years old range) and the wide range of participants in our study should be considered as a certain limitation.

Abnormalities of early exogenous ERPs such as N1 and P1 components can negatively affect endogenous potentials (e.g., N2 and P3) as well as response-locked potentials (ERN and Pe). The deficits in identification of distinct categories of stimuli at the early sensory stage result in a need to delegate task of differentiation of target stimulus from irrelevant one to the later, higher level of information processing stage. It is possible to suggest that individuals with ASD may engage some compensatory strategies necessary for successful target detection. 

## 5. Conclusions

The results of the study indicate an atypical manner of processing distracters and orienting attention to novelty in children with autism. The findings are in-keeping with our prior studies using different tasks with visual and auditory stimuli. Augmented early potentials and a delayed frontal P3a and parietal P3b to novel stimuli suggest low selectivity in pre-processing of distracters resulting in excessive information processing at later stages. This may indicate a reduction in the discriminative ability of the ASD group. These results may reflect a locally over-connected network where sensory inputs evoke abnormally large ERPs for unattended stimuli with signs of a reduction in selectivity. This may incur a high load at the later stages of perceptual and cognitive processing and response selection when novel distracter stimuli are differentiated from targets. 

## Figures and Tables

**Figure 1 behavsci-07-00079-f001:**
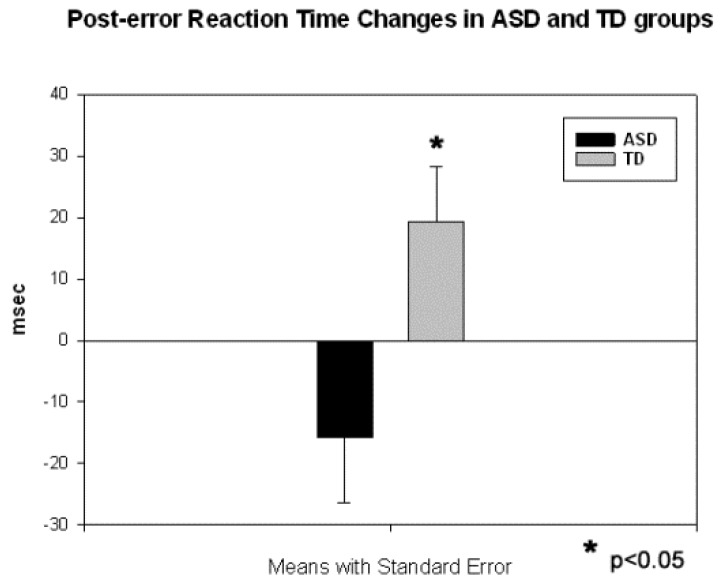
Post-error reaction time (RT) changes (calculated as a difference between the first post-error RT minus mean preceding RT) in ASD and TD children. Typical children show normative post-error RT slowing, conversely ASD children respond faster after having committed an error.

**Figure 2 behavsci-07-00079-f002:**
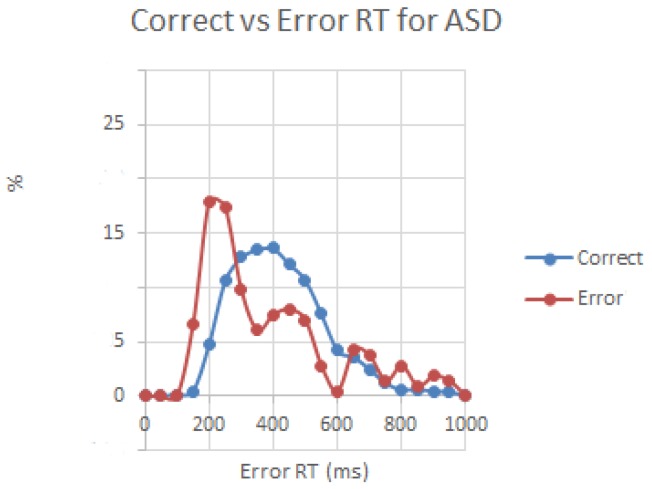
Distribution of RT in correct and error trials in children with ASD. Error trials had higher percentage in the faster bins of the histogram with moda around 200 ms vs. 400 ms in correct trials.

**Figure 3 behavsci-07-00079-f003:**
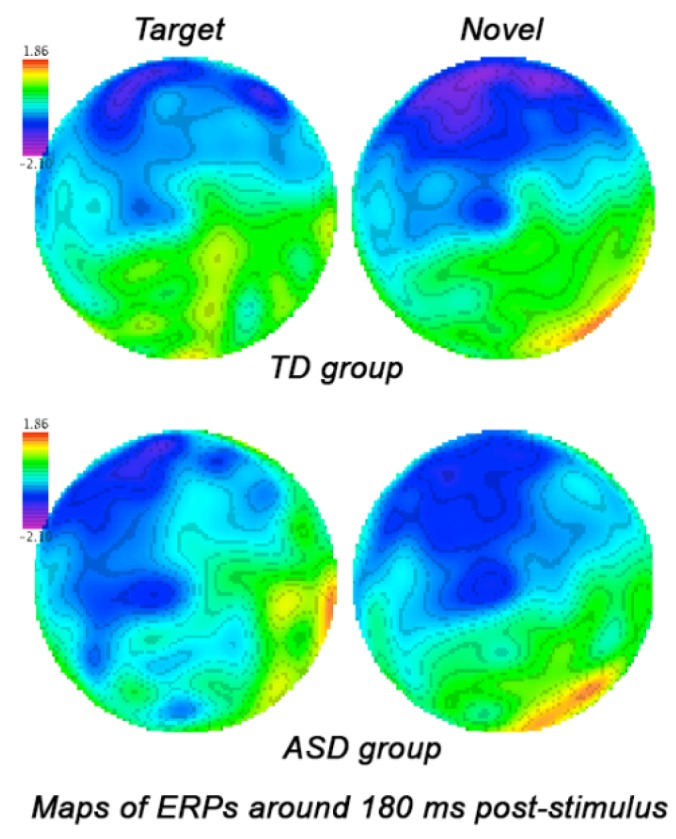
Grandaverage qEEG map of ERP to target and novel stimuli around 180 ms post-stimulus in ASD and TD children. Children with ASD showed comparable high negativity (N1 ERP component) to both targets and novels.

**Figure 4 behavsci-07-00079-f004:**
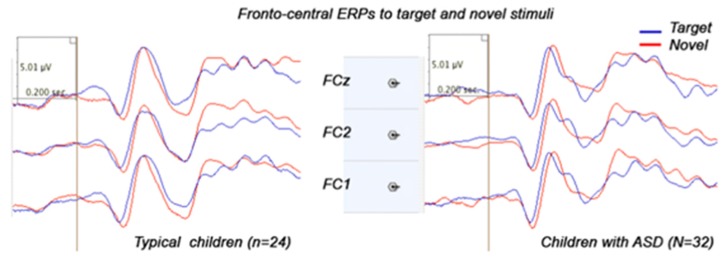
Screenshot of fronto-central ERPs to target and novel stimuli in ASD and TD children. The ASD children showed more negative N1, prolonger N2a and augmented P3a in response to unattended novel distracters.

**Figure 5 behavsci-07-00079-f005:**
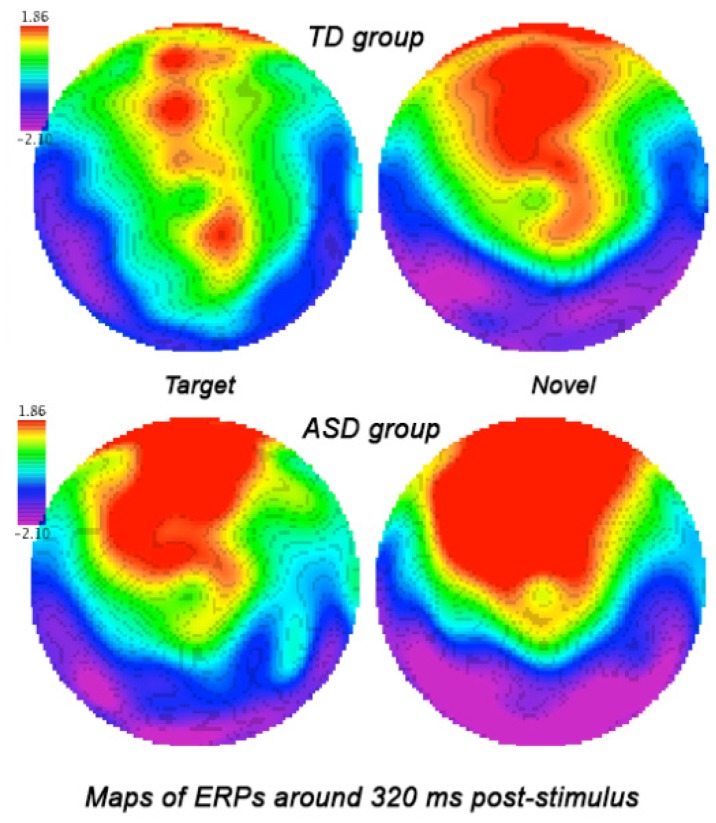
Grandaverage qEEG map of ERP to target and novel stimuli around 320 ms post-stimulus in ASD and TD children. Children with ASD showed higher positivity (P3a ERP component) to both targets and novels at the frontal and froto-central topographies.

**Figure 6 behavsci-07-00079-f006:**
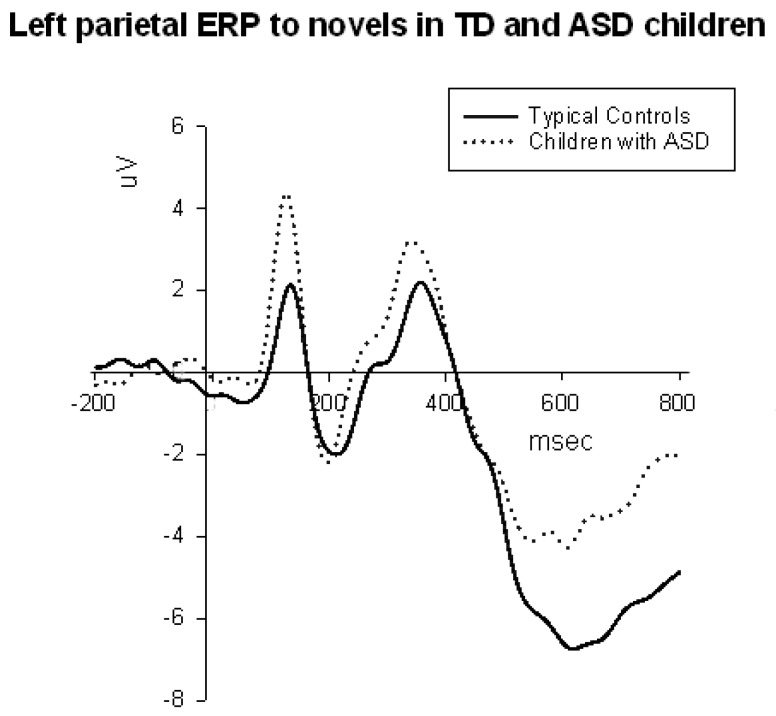
Left parietal ERPs to novels in ASD and TD children. The ASD group showed higher amplitude of P1 and P3b ERP components in response to novel distracters.

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
