# Peer review of "Atypical Processing of Novel Distracters in a Visual Oddball Task in Autism Spectrum Disorder"

_behavsci, 2017, doi:10.3390/bs7040079_

Round 1

Reviewer 1 Report

The paper ‘An ERP study of atypical processing of novel distracters in a visual oddball task in children with autism spectrum disorder’ is extremely interesting in his declination on P3 with a distractor. The research Design is appropriate and Method and Results are adequately described. The Discussion is very long and exhaustive.

Only Some little suggestions:   A smarter 'title' (e.g., You can delete 'an ERP study' and you can use not the term 'children' 

Lines 60-61: positive ERP components (delete peaks…)

Lines 78-81: this phrase is too long and not clear. Please, rephrase

Line 87: Missing reference to acronym TD. You use the explanation in the abstract (Typical Development) but you never explain in the introduction. Please, rewrite Typical Development (TD) children.

Line 95 Children with ASD and TD children… Kindly change in ‘ASD and TD children’ or rephrase.

Line 96 Could be better to use, according to ERP literature (i.e., Luck 2005) the terminology N2, P3 and so on… instead N200, P300 and so on…

Lines 213-221 Could be better add this section in section 2.1. ‘Participants’  (lines 103-130)

Line 234 ‘in children with ASD and TD children’ see comment Line 95 or You can change, in the captions the term ‘group’ ASD group and TD Group

Line 237: Fig. 2  Can You match and show the same Distribution of ASD with TD?

Line 289: Please conform the terminology in all the images: or you use TD children and ASD children or TD group and ASD group.

Line 309: same comment of line 289 Please conform the terminology: Typical Controls (???) the same: or TD group or….

Children with ASD: conform terminology

a general consideration: You use subjects with a range of age from 9 (with a mean of the sample of 13 years old) to 18 years old too, probably you can consider another synonymous which is closer to the idea of adolescent, some of whom approach even to older age (18 years old). This suggestion is motivated by the variability of ERP components, that is very different between children (term that usually includes subjects in a range between two and ten years old) , adolescents and adults (e.g.,  Tamnes, C. K., Walhovd, K. B., Torstveit, M., Sells, V. T. & Fjell, A. M. Performance monitoring in children and adolescents: A review of developmental changes in the error-related negativity and brain maturation. Developmental Cognitive Neuroscience 6, 1–13 (2013).

Author Response

The paper ‘An ERP study of atypical processing of novel distracters in a visual oddball task in children with autism spectrum disorder’ is extremely interesting in his declination on P3 with a distractor. The research Design is appropriate and Method and Results are adequately described. The Discussion is very long and exhaustive.

    -Thank you for the general positive evaluation of our manuscript.

Only some little suggestions:   A smarter 'title' (e.g., You can delete 'an ERP study' and you can

use not the term 'children' 

    -Following recommendation of Reviewer 1 we dropped “an ERP study” in the title and do not use term “children” in the title.

Lines 60-61: positive ERP components (delete peaks…)

    - Changed per reviewer’s comment

Lines 78-81: this phrase is too long and not clear. Please, rephrase

    -The phrase(s) were redone and shortened as suggested.

Line 87: Missing reference to acronym TD. You use the explanation in the abstract (Typical Development) but you never explain in the introduction. Please, rewrite Typical Development (TD) children.

    -An explanation for the abbreviation of TD was inserted as requested

Line 95 Children with ASD and TD children… Kindly change in ‘ASD and TD children’ or rephrase.

     -Changed as recommended

Line 96 Could be better to use, according to ERP literature (i.e., Luck 2005) the terminology N2, P3 and so on… instead N200, P300 and so on…

      -Following recommendation of reviewer for the whole text we started using N1, N2 and P3 (P3a, P3b) terminology instead of N100, N200 and P300..

Lines 213-221 Could be better add this section in section 2.1. ‘Participants’  (lines 103-130)

    -The 3.1. section was moved to section 2.1. Numeration in section 3 was adjusted accordingly.

Line 234 ‘in children with ASD and TD children’ see comment Line 95 or You can change, in the captions the term ‘group’ ASD group and TD Group

    - Adjusted in a similar manner as at line 95.

Line 237: Fig. 2  Can You match and show the same Distribution of ASD with TD?

-       We have some figure drafts for such distribution, for instance for errors between ASD and TD, but we decided not include them as a major point of the study was our focus on ERP measures rather than behavioral responses comparisons.

Line 289: Please conform the terminology in all the images: or you use TD children and ASD children or TD group and ASD group.

-       We standardized terminology in all our figures.

Line 309: same comment of line 289 Please conform the terminology: Typical Controls (???) the same: or TD group or….

Children with ASD: conform terminology

-       We tried to conform the terminology across the whole paper in our revised version.

However, in Results section where we conducted statistical analysis to find between group differences it seemed more logical to use AD and TD groups rather than ASD and TD children.

A general consideration: You use subjects with a range of age from 9 (with a mean of the sample of 13 years old) to 18 years old too, probably you can consider another synonymous which is closer to the idea of adolescent, some of whom approach even to older age (18 years old). This suggestion is motivated by the variability of ERP components, that is very different between children (term that usually includes subjects in a range between two and ten years old),adolescents and adults (e.g.,  Tamnes, C. K., Walhovd, K. B., Torstveit, M., Sells, V. T. & Fjell, A. M. (2013) Performance monitoring in children and adolescents: A review of developmental changes in the error-related negativity and brain maturation. Developmental Cognitive Neuroscience,  6, 1–13.

-        There was an added statement that ERP variability (especially ERN variability) might be dependent on the age of children (specifically differences between children 9-12 yrs old and adolescents—teenagers in 12-18 years old range) and the reference of Tamnes et al., 2013 was added as [72]. This was acknowledged as a limitation. In general, referring to individuals up to 21 yrs as children is formally in agreement with the NIH definition of children. We do understand, nevertheless, that there are significant normative developmental differences between younger children and adolescents.

Reviewer 2 Report

The current manuscript describes results of an oddball P300 paradigm with children and adolescents with and without autism. Results suggest significant between group differences, and have implications for early sensory processing and sensory gating in autism. 

The manuscript is well-written, and the results are clearly stated. I have a few minor concerns, but overall feel as though this paper requires only minor revisions. The concerns I have are noted below, in the order they appear. 

The manuscript refers to individuals on the autism spectrum as "autistic individuals" in several places. It is often preferred to use person-centered language such as, "individuals with autism".  I encourage the authors to make the language consistent throughout the manuscript, and to utilize person-centered language. 

In section 3.1 (participants), the authors describe how IQ information was obtained for individuals with autism, but do not mention typically developing individuals. I assume this means that TD participants were not tested, but it would help to clarify this point in the manuscript. 

I believe the caption for Figure 1 has a typo, as it reads "...children with ASD on opposite respond faster after committed error". I believe the authors mean something along the lines of "...conversely, children with ASD respond faster after having committed an error". 

This manuscript is comprehensive, and will be helpful for other researchers engaging in neuroscience studies of orienting, sensory gating, and habitation in ASD. 

Author Response

Comments and Suggestions for Authors

The current manuscript describes results of an oddball P300 paradigm with children and adolescents with and without autism. Results suggest significant between group differences, and have implications for early sensory processing and sensory gating in autism. 

    The manuscript is well-written, and the results are clearly stated. I have a few minor concerns, but overall feel as though this paper requires only minor revisions. The concerns I have are noted below, in the order they appear. 

-       Thank you for the positive evaluation of the style of the manuscript and it’s significance.

The manuscript refers to individuals on the autism spectrum as "autistic individuals" in several places. It is often preferred to use person-centered language such as, "individuals with autism".  I encourage the authors to make the language consistent throughout the manuscript, and to utilize person-centered language. 

-       We agree with the comment of reviewer 2 and replaced the term “autistic individuals” with “individuals with autism” as more appropriate one.

In section 3.1 (participants), the authors describe how IQ information was obtained for individuals with autism, but do not mention typically developing individuals. I assume this means that TD participants were not tested, but it would help to clarify this point in the manuscript. 

    -Yes, we were not able to collect IQ information from all our typically developing individuals, so we now acknowledge that fact in the revised version of the manuscript.  Testing all TD children using WIC or WASI was not technically feasible due to our procedure of recruitment of control subjects. We understand that this was a limitation.

I believe the caption for Figure 1 has a typo, as it reads "...children with ASD on opposite respond faster after committed error". I believe the authors mean something along the lines of "...conversely, children with ASD respond faster after having committed an error". 

-       Thank you for the suggestion, we changed legend to Fig. 1 as recommended.

This manuscript is comprehensive, and will be helpful for other researchers engaging in neuroscience studies of orienting, sensory gating, and habitation in ASD. 

-        Thank you for the positive comments regarding the significance of our manuscript.